# ECMNet: Lightweight Semantic Segmentation with Efficient CNN-Mamba Network

## Abstract

In the past decade, Convolutional Neural Networks (CNNs) and Transformers have achieved wide application in semantic segmentation tasks. Although CNNs with Transformer models greatly improve performance, the global context modeling remains inadequate. Recently, Mamba achieved great potential in vision tasks, showing its advantages in modeling long-range dependency. In this paper, we propose a lightweight Efficient CNN-Mamba Network for semantic segmentation, dubbed as ECMNet. ECMNet combines CNN with Mamba skillfully in a capsule-based framework to address their complementary weaknesses. Specifically, We design a Enhanced Dual-Attention Block (EDAB) for lightweight bottleneck. In order to improve the representations ability of feature, We devise a Multi-Scale Attention Unit (MSAU) to integrate multi-scale feature aggregation, spatial aggregation and channel aggregation. Moreover, a Mamba enhanced Feature Fusion Module (FFM) merges diverse level feature, significantly enhancing segmented accuracy. Extensive experiments on two representative datasets demonstrate that the proposed model excels in accuracy and efficiency balance, achieving 70.6% mIoU on Cityscapes and 73.6% mIoU on CamVid test datasets, with 0.87M parameters and 8.27G FLOPs on a single RTX 3090 GPU platform.

## 1 Introduction

Semantic segmentation aims to assign a label to each pixel in a given image, which is widely applied in autonomous driving(Sanchez et al., 2025), remote sensing(Jing et al., 2025), and agriculture(Luo et al., 2024), and more.

Early semantic segmentation primarily relied on CNNs, employing techniques like large convolutional kernels (Peng et al., 2017), dilated convolutions(Chen et al., 2017), and feature pyramids(Zhao et al., 2017) to extend receptive fields. However, these CNN-based approaches remained limited in capturing long-range dependencies. The advent of Transformers(Vaswani et al., 2017) enabled more effective global context modeling in subsequent segmentation methods. Learning global context dependencies is essential for extracting global semantic features, particularly in intensive tasks like semantic segmentation. The rise of Visual Transformer (ViT)(Dosovitskiy et al., 2020) has injected a new paradigm for semantic segmentation. SETR(Zheng et al., 2021) slices images into sequences for the first time and captures global context feature through a self-attentive mechanism, outperforming traditional CNN models on complex scene datasets such as Cityscapes. Meanwhile, SegFormer(Xie et al., 2021) further optimized the architectural design by proposing a hierarchical Transformer encoder with a lightweight MLP decoder to achieve multi-scale feature fusion. However, the square-level computational complexity of Transformer limits its application to high-resolution images with insufficient sensitivity to local details.

To tackle the limitation of the above single model and extract fine spatial details, some models treated semantic segmentation tasks by integrating CNN with Transformer. For instance, HResFormer(Ren & Li, 2025), PFormer(Gao et al., 2025), and DMFC-UFormer(Garbaz et al., 2025) have achieved satisfactory results in the field of medical image segmentation. However, the self-attention mechanism in CNN-Transformer methods still poses challenges in terms of speed and memory usage when dealing with long-range visual dependencies, especially processing high-resolution images.

Unlike previous Transformer, Mamba(Gu & Dao, 2023) shows great potential for high-resolution image by efficient sequence modeling with linear complexity. Vision Mamba(Liu et al., 2024b)

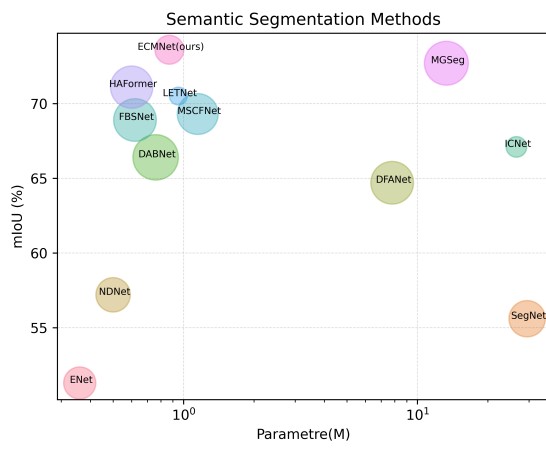

Figure 1: Accuracy and model parameters comparison of ECMNet and other lightweight models on CamVid dataset. A larger circle denotes a faster inference speed.

have recently demonstrated remarkable success in various computer vision tasks. For example, in the field of 3D medical imaging, SegMamba(Xing et al., 2024) achieves real-time inference on the colorectal cancer dataset CRC-500, with a speedup of 30% compared to 3D UNet. In addition, CM-UNet(Liu et al., 2024a) introduces a Mamba decoder into a CNN encoder to bridge local and global features through a channel-space attention mechanism, achieving higher mIoU on the ISPRS Vaihingen dataset.

To accommodate limited computational resources and mobile device application, lightweight semantic segmentation models receive higher attention. For example, LEDNet(Wang et al., 2019) employed channel split-and-shuffle operations within residual blocks, significantly lowering computational complexity. While CFPNet(Ding et al., 2024) designed Channel-wise Feature Pyramid (CFP) module to significantly reduces model parameters and model scale by extracting various level feature map and contextual feature information jointly. LETNet(Xu et al., 2023) used an LDB module and FE module for enhanced efficiency and accuracy with reduced model complexity.

Motivated by the success of Mamba and lightweight approaches in semantic segmentation tasks, We propose ECMNet, an efficient CNN-Mamba hybrid network for lightweight semantic segmentation, optimized for minimizing model size and computational requirements. As depicted in Figure 1, the proposed ECMNet achieves a excellent balance between the accuracy, inference speed of the model, and model parameters. The main contributions of our paper are four folds:

- We firstly propose a novel lightweight Efficient CNN-Mamba Network (EMCNet) for for semantic segmentation. ECMNet utilizes U-shape encoder-decoder structure as backbone and regards the Feature Fusion Module (FFM) as a capsule network to capture global context information. Specially, FFM introduces SS2D block, a variant of Mamba, to learn long-range dependencies.

- We design a lightweight Enhanced Dual-Attention Block (EDAB) to extract multi-dimensional semantic information. EDAB consists of Dual-Direction Attention(DDA), Channel Attention (CA) and various convolution modules, realizing less model parameters and computational quantities.

- We develop a Multi-Scale Attention Unit (MSAU) to improve the representations ability of feature, which further refines the local details and global contextual information.

- ECMNet achieved 70.6% mIoU on the Cityscapes dataset on the single RTX 3090 GPU with only 0.87M of parameters, realizing the better trade-off between performance and parameters. Meanwhile, our proposed method achieved 73.6% of the highest performance on the CamVid dataset, which demonstrates the effectiveness and generalization of our proposed ECMNet.

## 2 RELATED WORK

### 2.1 SEMANTIC SEGMENTATION METHODS BASED ON CNN AND TRANSFORMER

Due to the efficient local feature representation capabilities of CNNs, semantic segmentation has also advanced tremendously. Following the revolutionary CNN, FCN and U-Net, many new architectures have been refined on this basic principle. However, CNN-based methods face issues with the trade-off between image resolution and a limited receptive field. To address these challenges, DeepLab and PSPNet build atrous spatial pyramid pooling using atrous convolutions in parallel way, which better utilizes various level contextual feature information.

Additionally, researchers has drawn much interest in self-attention mechanism because of the advantages in modeling feature dependencies. ECANet(Wang et al., 2020b) introduced a local cross-channel interaction mechanism that operates without dimensionality reduction and an adaptive selection method for determining optimal kernel sizes in one-dimensional convolutions In addition, DANet(Fu et al., 2019) employed ResNet as its backbone architecture, integrating parallel attention modules in both spatial and channel dimensions. This design effectively captures long-range feature dependencies, enhancing segmentation performance.

However, all these methods build an enormous computational challenge for the machine. So the lightweight semantic segmentation networks were proposed. For example, ICNet(Zhao et al., 2018) captured high-level semantic information and low-level spatial details by utilizing multi-scale images as input. BiseNet(Zhao et al., 2022) and BiseNet-v2(Yu et al., 2021) introduced two-path architecture, which is responsible for providing detailed information supplement and extracting deep semantic information respectively. Furthermore, in order to enhance the feature expression and reduce the computation, the point-wise attention were designed. ESPNet(Mehta et al., 2018) and ESPNet-v2(Lin et al., 2023) fused decomposed convolution into point-wise convolution, which greatly reduces the number of parameters and computation. In addition, LEDNet(Wang et al., 2019) introduced an attention pyramid network in its decoder, effectively reducing overall model complexity while maintaining performance. To alleviate the limitation of the single model and extract fine spatial details, some method combined CNN with Transformer came into being. For instance, LETNet(Xu et al., 2023) incorporated two key components: a lightweight dilated bottleneck module and an enhanced feature refinement module combining CNN-Transformer capsules. This architecture can capture long-range feature dependencies for better segmentation results. HAFormer(Xu et al., 2024a) integrates CNN-based hierarchical feature learning and Transformer-base global context modeling to further capture global representation. While existing methods still have room for enhancement in global feature representation and complexity.

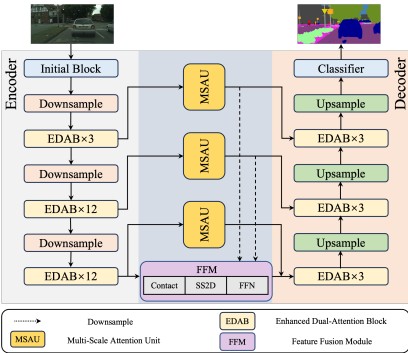

Figure 2: The overall network architecture of Efficient CNN-Mamba Network (EMCNet)

### 2.2 SEMANTIC SEGMENTATION METHODS BASED ON MAMBA

Mamba(Gu & Dao, 2023) has achieved great success because of sequence modeling with linear complexity. Meanwhile, Vision Mamba(Liu et al., 2024b) have recently proved once again the possibilities in the field of computer vision tasks, especially in semantic segmentation. For example, CM-UNet(Liu et al., 2024a) built the core segmentation decoder by employing channel and spatial

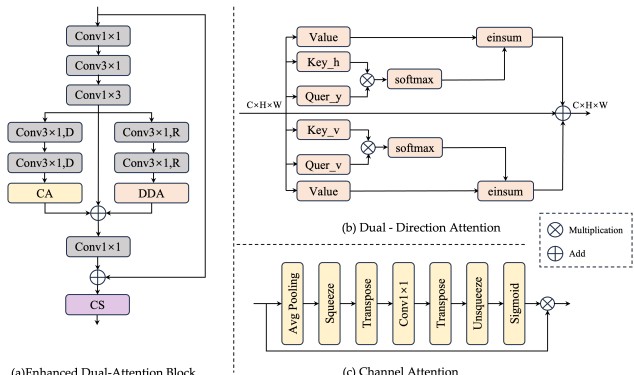

Figure 3: The overall architecture of the proposed Enhanced Dual-Attention Block (EDAB)

attention as the gate activation condition of the vanilla Mamba, which enhances the feature interaction and global-local information fusion. RS³Mamba(Ma et al., 2024) utilized VSS blocks(Gu & Dao, 2023) to acheive better segmentation in remote sensing by constructing an auxiliary branch to enhance a convolution-based main branch. Sigma(Wan et al., 2025) intorduced two novel modules, a Siamese encoder and a Mamba-based fusion mechanism, to achieving global receptive fields with linear complexity.

Although he above methods have achieved good results, the current Mamba-based segmentation methods applied to remote sensing without considering the model size, which results in large model parameters. In addition,a one-dimensional sequence input of Mamba in the image domain disrupted local structural relationships with global context information. On the other hand, the absence of fine-grained local features results in imprecise segmentation, CNN architectures effectively compensate by preserving spatial details through local feature extraction. In order to better handle semantic segmentation task and reduce model parameters, we tend to explore a novel lightweight method integrating CNN with Mamba.

## 3 PROPOSED METHOD

### 3.1 OVERALL NETWORK ARCHITECTURE

As shown in Figure 2, the overall network architecture of our proposed EMCNet consists of four components: a CNN encoder improved with enhanced dual-attention blocks, a CNN decoder with subtle difference from encoder, an efficient Mamba-based feature fusion module and three long skip connections enhanced with multi-scale attention unit. Specifically, the CNN-based encoder-decoder architecture extracts localized features for detailed spatial representation. The Mamba-based FFM can capture complex spatial information and long-range feature dependencies by state space model (SSM) to optimize global feature representations and computational complexity. The three long-distance skip connections generate more high-quality segmentation by focusing on low-level spatial information and high-level semantic information respectively. The above mentioned elaborated modules make it more efficient for ECMNet to fully integrate local and global feature information.

### 3.2 ENHANCED DUAL-ATTENTION BLOCK

As shown in Figure 3, the structure of EDAB is inspired the idea of multi-head attention mechanism. The module is designed to focus different level feature information and keep network parameters as few as possible. Firstly, the input feature passes through a bottleneck structure that utilizes a 1×1 convolution to reduce the number of channels to half, significantly reducing the computational complexity and the number of parameters. Obviously, this will sacrifice a part of the accuracy,it will be more beneficial to introduce 3×1 convolution and 1×3 convolution more than make up for the loss at this point. Meanwhile, the two decomposed convolutions not only obtain a wider respective field for capturing a larger range of contextual feature information but also consider the model parameters

and calculation complexity. The core of EDAB lies in its two-branch path, which captures local and global feature information respectively. Decompose convolution in one branch processes local and short-distance feature information, complemented by atrous convolution in the parallel branch for global feature integration. Then, the channel contains most of the feature information and the spatial feature information is key to enhance performance and suppress noise interference. Therefore, the two branches utilize Channel Attention (CA) and Dual-Direction Attention (DDA), which aims to build different attention matrix to learn multi-dimensional feature information and improve feature expression. Finally, the outputs from both designed pathes and intermediate features are integrated and processed by a 1×1 point-wise convolution to restore the original channel dimensionality. A channel shuffle strategy is applied at the end of EDBA to establish inter-channel correlations and overcome information fragmentation The detail operation is shown as follows:

$$F_{up\_branch} = Conv_{1 \times 3}(Conv_{3 \times 1}(Conv_{1 \times 1}(x))), \tag{1}$$

$$F_{mid\_branch\_1} = Conv_{CA}(Conv_{1 \times 3, D}(Conv_{3 \times 1, D}(F_{up\_branch}))), \tag{2}$$

$$F_{mid\_branch\_2} = Conv_{DDA}(Conv_{1 \times 3, D, R}(Conv_{3 \times 1, D, R}(F_{up\_branch}))), \tag{3}$$

$$Y_1 = Conv_{CS}(f_{1 \times 1}(F_{up\_branch} + F_{mid\_branch\_1} + F_{mid\_branch\_2}) + x), \tag{4}$$

where $x$ denotes the input of the EDAB, $Y_1$ denotes the output feature map of the EDAB, and $Conv_{k \times k}(\cdot)$ are normal convolution operation. Among the suffix, D denotes depth-wise convolution, R is the atrous rates of atrous convolution, CA represents Channel Attention, DDA represents Dual-Direction Attention and CS denotes the shuffle operation of channel.

## 3.3 MULTI-SCALE ATTENTION UNIT

On the one hand, lower layers preserve fine spatial details with limited semantics, on the other hand, higher layers offer strong semantic representation at lower spatial resolution. Therefore, it is an efficient strategy to combine the low-level rich spatial information and high-level rich semantic information for semantic segmentation tasks. Inspired by U-Net, we use the same resolution connections to integrate the high-level feature maps and low-level feature maps. In order to better process the three long connections, we design a Multi-Scale Attention Unit (MSAU) to enhance the ability of feature representation. As shown in Figure 4, MSAU is carried out from two branches, one is the Multi-Scale Spatial Aggregation, the other is the Channel Aggregation.

In the Multi-Scale Spatial Aggregation, the input feature map is utilized 1×1 convolution to convert from C channel to C/2 channel. In order to reduces the amount of parameter and computation while retaining the ability of multi-scale feature extraction, the next feature map goes through different sizes of depth-separable convolution,such as 3×3, 5×5 and 7×7. Meanwhile, the outputs of different scale convolutions obtain multi-scale feature information enhancing the multi-scale perception capability of the model. Then, the multi-scale fused feature map compresses the height dimension to 1 by adaptive average pooling, and generates a spatial attention map by 7×7 depth separable convolution, 1×1 convolution and Sigmoid activation function. At the same time, by multiplying with the multi-scale fused feature map, the processed feature highlights the key spatial regions and suppresses the irrelevant information. At last, the channel of model is converted from C/2 back to C by using 1x1 convolution, and the attention map reflects the importance of the different locations of feature map. For channel aggregation, the input feature map uses average pooling and maximum pooling to obtain average channel features and maximum channel features respectively, which captures channel statistics from different angles. The MSAU multiplies the spatial and channel aggregation results and adds them with the original input feature maps to obtain the output feature maps.

This design allows the MSAU module to fused the low-level spatial information to the high-level semantic information more effectively, and further enhance the ability of feature expression The

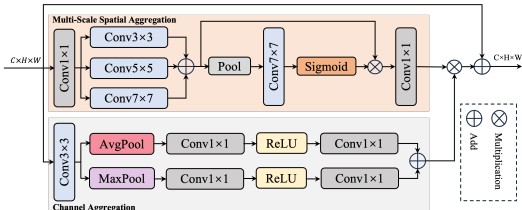

Figure 4: The architecture of our proposed Multi-Scale Attention Unit (MASU)

detail operation can be defined as:

$$X_1 = Conv_{(3\times3)}(Conv_{(1\times1)}(x)) + Conv_{(5\times5)}(Conv_{(1\times1)}(x))$$
$$+Conv_{(7\times7)}(Conv_{(1\times1)}(x)) \tag{5}$$
$$X_2 = Conv_{(1\times1)}(X_1 \otimes Sigmoid(Conv_{(7\times7)}(Pool(x)))) \tag{6}$$
$$X_3 = Conv_{(1\times1)}(ReLU(Conv_{(1\times1)}(AvgPool(Conv_{(3\times3)}(x))))) \tag{7}$$
$$X_4 = Conv_{(1\times1)}(ReLU(Conv_{(1\times1)}(MaxPool(Conv_{(3\times3)}(x))))) \tag{8}$$
$$Y_2 = x + (X_2 \otimes (X_3 + X_4)) \tag{9}$$

where $x$ denotes the input of the MSAU and $Y_2$ represents the output feature map of the MSAU. Among the formulas, $Conv_{k\times k}(\cdot)$ are normal convolution operation. $\otimes$ denotes element-by-element multiplication, $Pool(\cdot)$ denotes the adaptive average pooling, $AvgPool(\cdot)$ is average pooling, $MaxPool(\cdot)$ is maximum pooling, $ReLU(\cdot)$ is rectified linear unit and $Sigmoid(\cdot)$ is the Sigmoid activation function.

## 3.4 Feature Fusion Module

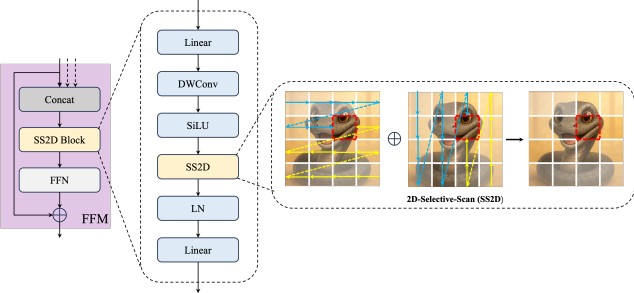

Figure 5: The architecture of our proposed Feature Fusion Module (FFM)

Motivated by by the effectiveness of Mamba in linear-complexity sequence modeling, we design a Feature Fusion Module (FFM) by introduce 2D-Selective-Scan (SS2D) block for better capturing global representations with less network parameters and computational quantities. As shown Figure 5, the FFM enriches the feature diversity by integrating different scale feature information from the multi-level the MSAU and the encoder through the concatenation operation. Then, the SS2D block further extracts and fuses the features through a series of linear transformations and 2D convolution operations, which employs a selective scanning mechanism to enhance the feature representation ability. Finally, Feed-Forward Network (FFN) performs a nonlinear transformation to adjust the weight distribution of features, highlighting the key features and suppressing the redundant information, so as to improve performance in handling complex tasks. The designed FFM can effectively fuse multi-scale features and capture both local detail information and overall semantic features, great improving the performance of the model in semantic segmentation tasks. The complete operation is shown as follows:

$$X_{FFN} = FFN(SS2D(Concat(x_{encoder}, x_{MSAU1}, x_{MSAU2}))) \tag{10}$$
$$Y_3 = X_{FFN} + x_{encoder} \tag{11}$$

where $x_{encoder}, x_{MSAU1}, x_{MSAU2}$ denotes the out of the Encoder and MSAU resectively, $Y_3$ denotes the output feature map of the FFM. Among the formulas, $Concat(\cdot)$ is normal concatenation operation. $SS2D(\cdot)$ is the 2D-Selective-Scan block and $FFN(\cdot)$ is the eed-Forward Network.

## 4 EXPERIMENTS

### 4.1 DATASETS

- **Cityscapes.** This dataset is composed of high-quality 5,000 images, annotated at the pixel level. The images are primarily scenes of driving within urban settings, captured across 50 different cities with a resolution of 2,048×1,024. The dataset was divided into training sets(2,975 images), validation sets (500 images), and test sets (1,525 images)

- **CamVid.** The CamVid dataset, developed by the University of Cambridge, contains urban road scene images captured from a driving perspective (960×720 resolution). Its 700+ annotated samples support supervised learning, featuring 11 representative object classes that effectively capture urban road elements. This diversity in objects and well-annotated classes makes it particularly suitable for our segmentation accuracy research.

### 4.2 IMPLEMENTATION DETAILS

Our proposed ECMNet, implemented in PyTorch, was trained using an NVIDIA RTX 3090 GPU. We employ random initialization and full training from scratch, extending the maximum epoch count to 1,000 In training the ECMNet model, different parameter configurations are used for the Cityscapes and the CamVid datasets. The parameter configurations for the Cityscapes dataset included a batch size of 6, the use of a cross-entropy loss function, a stochastic gradient descent (SGD) optimizer with a momentum of 0.9, a weight decay of $1e-4$, an initial learning rate of 0.045, and the use of a polynomial learning rate strategy. The parameter configuration for the CamVid dataset included a batch size of 8, the same cross-entropy loss function, the Adam optimizer, a momentum of 0.9, a weight decay of 0.0002, an initial learning rate of $1e-3$, and a polynomial learning rate strategy as well. These parameters are set to adapt to the characteristics of different datasets with a view to obtaining the best training results.

### 4.3 ABLATION STUDIES

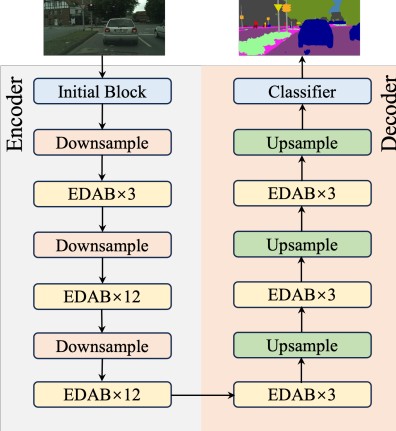

Figure 6: The simple structure of the baseline model

We design a series of ablation experiments to validate the effectiveness of each module in our proposed model. As shown in Figure 6, The baseline model used for comparison is structured as simple U-shape type, including the standard Encoder and Decoder. The Encoder and Decoder consist of multiple lightweight enhanced dual-attention blocks (EDABs), which are modeled to achieve an average mIoU of 69.92% on the Camvid validation set.

In the long connection ablation experiments (A Group), the effect of gradually adding Line 1, Line 2 , and Line 3 is investigated. The observed 0.61% enhancement after adding Line 1 substantiates that shallow information effectively aids semantic feature information reconstruction. Meanwhile, With three long skip connections, the model achieves a 1.29% mIoU enhancement. These results

further demonstrate the significance of long-range skip connections for semantic segmentation task. In the MSAU ablation experiments (B group), the MSAU module is added gradually in the long connection. A comparison between B1 and A1 reveals that adding the MSAU module to long connections only adds 9.43K parameters, but improves the performance by 0.92% of mIoU. In the last ablation experiments (C Group), the introduction of the Feature Fusion Module (FFM) improves the performance of the model by 1.11% of mIoU. Finally, as the finalized architecture (C3), our proposed ECMNet improves performance by 3.7% mIoU compared to the baseline model. All the above experiments shown in Table 1 fully validate the efficacy of our proposed modules and strategies

Table 1: Extensive ablation study for the proposed ECMNet on Camvid dataset.

| Architecture | Method | | | | | | | Parameter (K)↓ | FLOPs (G)↓ | mIoU (%)↑ |
| | Long Connection | | | MSAU | | | FFM | | | |
| | 1 | 2 | 3 | 1 | 2 | 3 | | | | |
|---|---|---|---|---|---|---|---|---|---|---|
| **Baseline** | - | - | - | - | - | - | - | 775.57 | 7.56 | 69.92 |
| **A1** | ✓ | - | - | - | - | - | - | 777.93 | 7.57 | $70.53^{0.61\uparrow}$ |
| **A2** | ✓ | ✓ | - | - | - | - | - | 796.41 | 7.64 | $70.92^{1.00\uparrow}$ |
| **A3** | ✓ | ✓ | ✓ | - | - | - | - | 805.67 | 7.79 | $71.21^{1.29\uparrow}$ |
| **B1** | - | - | - | ✓ | - | - | - | 787.34 | 7.57 | $71.45^{1.53\uparrow}$ |
| **B2** | - | - | - | ✓ | ✓ | - | - | 805.82 | 7.67 | $72.65^{2.73\uparrow}$ |
| **B3** | - | - | - | ✓ | ✓ | ✓ | - | 815.08 | 7.90 | $73.22^{3.30\uparrow}$ |
| **C1** | - | - | - | - | - | - | ✓ | 827.80 | 7.80 | $70.75^{0.83\uparrow}$ |
| **C2** | ✓ | ✓ | ✓ | - | - | - | ✓ | 863.93 | 8.06 | $71.03^{1.11\uparrow}$ |
| **C3 (ours)** | ✓ | ✓ | ✓ | ✓ | ✓ | ✓ | ✓ | 871.11 | 8.27 | $\mathbf{73.62}^{3.70\uparrow}$ |

A, B, C denote the long connection, the feature enhancement and the feature fusion respectively. No. of A, B and C denotes the the stack of same or different modules.

Table 2: Performance comparison of our proposed ECMNet and other representative methods on the Cityscapes dataset.

| Method | Year | Resolution (width×height) | Backbone | Parameter (M)↓ | FLOPs (G)↓ | Speed (FPS)↑ | mIoU (%)↑ |
|---|---|---|---|---|---|---|---|
| SegNet(Badrinarayanan et al., 2017) | 2017 | 640×360 | VGG-16 | 29.50 | 286.0 | 17 | 57.0 |
| ENet(Paszke et al., 2016) | 2016 | 512×1024 | No | 0.36 | 3.8 | 135 | 58.3 |
| ESPNet(Mehta et al., 2018) | 2018 | 512×1024 | ESPNet | 0.36 | – | 113 | 60.3 |
| NDNet(Yang et al., 2020) | 2021 | 512×1024 | No | 0.50 | 3.5 | 120 | 61.1 |
| CGNet(Wu et al., 2020) | 2021 | 360×640 | No | 0.49 | 6.0 | – | 64.8 |
| ADSCNet(Wang et al., 2020a) | 2020 | 512×1024 | No | – | – | 77 | 67.5 |
| ERFNet(Romera et al., 2017) | 2017 | 512×1024 | No | 2.10 | – | 42 | 68.0 |
| BiseNet-v1(Zhao et al., 2022) | 2018 | 768×1536 | Xception | 5.80 | 14.8 | 106 | 68.4 |
| ICNet(Zhao et al., 2018) | 2018 | 1024×1024 | PSPNet-50 | 26.50 | 28.3 | 30 | 69.5 |
| DABNet(Li et al., 2019a) | 2019 | 1024×2048 | No | 0.76 | 42.4 | 28 | 70.1 |
| CFPNet(Ding et al., 2024) | 2021 | 1045×2048 | No | 0.55 | – | 30 | 70.1 |
| FPENet(Liu & Yin, 2019) | 2019 | 512×1024 | No | 0.40 | 12.8 | 55 | 70.1 |
| LEDNet(Wang et al., 2019) | 2019 | 512×1024 | No | 0.94 | – | 40 | 70.6 |
| DFANet(Li et al., 2019b) | 2019 | 1024×1024 | Xception | 7.80 | 3.4 | 100 | 71.3 |
| STDC1-50(Fan et al., 2021) | 2021 | 512×1024 | – | 8.40 | – | 87 | 71.9 |
| SegFormer(Xie et al., 2021) | 2021 | 512×1024 | MiT-B0 | 3.80 | 17.7 | 48 | 71.9 |
| MSCFNet(Gao et al., 2021) | 2022 | 512×1024 | No | 1.15 | 17.1 | 50 | 71.9 |
| FPANet(Wu et al., 2022) | 2022 | 512×1024 | – | 14.10 | – | – | 72.0 |
| MLFNet(Fan et al., 2022) | 2023 | 512×1024 | ResNet-34 | 13.03 | 15.5 | 72 | 72.1 |
| BiseNet-v2(Yu et al., 2021) | 2021 | 512×1024 | Xception | 3.40 | 21.2 | 156 | 72.6 |
| MGSeg(He et al., 2021) | 2021 | 1024×1024 | ShuffleNet-v2 | 4.50 | 16.2 | 101 | 72.7 |
| PCNet(Lv et al., 2021) | 2022 | 1024×2048 | Scratch | 3.40 | 21.2 | 156 | 72.6 |
| LETNet(Xu et al., 2023) | 2023 | 512×1024 | No | 0.95 | 13.6 | 150 | 72.8 |
| SCTNet-S(Xu et al., 2024b) | 2024 | 512×1024 | No | 4.6 | 451.2 | 160.3 | 72.8 |
| HSB-Net(Li et al., 2021) | 2021 | 512×1024 | ResNet-34 | 12.10 | – | 124 | 73.1 |
| LBN-AA(Dong et al., 2020) | 2021 | 448×896 | No | 6.20 | 49.5 | 51 | 73.6 |
| **ECMNet (Ours)** | – | 1024×1024 | No | 0.87 | 8.27 | 43 | **70.6** |

The gray box denotes the best value of the current metric.

## 4.4 COMPARISONS WITH SOTA METHODS

In this section, we compare state-of-the-art semantic segmentation methods in recent years on the Cityscapes and CamVid datasets to verify that our approach achieves a better balance between performance and parameters. Our evaluation is based on three key metrics: model parameters, floating-point operations (FLOPs) and mIoU.

**Evaluation Results on Cityscapes Dataset.** As shown in Table 2, the model with a larger number of parameters and computation obviously achieves excellent segmentation results. However, computational complexity of model is high and its operation speed is slow, which is unsuitable for real-time intelligent embedded devices. In contrast, lightweight models like NDNet(Yang et al., 2020), CGNet(Wu et al., 2020), CFPNet(Ding et al., 2024) , LEDNet(Wang et al., 2019) and LET-Net(Xu et al., 2023) are computationally efficient, which lack overall performance, especially in accuracy. Obviously, the LBN-AA(Dong et al., 2020) achieved the highest mIoU with 6.2M model parameters which are far more than our proposed approach. Meanwhile, the ESPNet(Mehta et al., 2018) utilized the least parameters to realize 60.3% mIoU, which is significantly lower than the performance of our method. Our proposed ECMNet only with a 0.87M parameters achieved a higher 70.6% mIoU. In Figure 7 of Appendix, we also show the visualization outcomes of these methods on the Cityscapes. Our proposed ECMNet can get better segmentation results with less model parameters, which benefits from well-design structure, and the utilization of the Mamba. These results fully demonstrate that our proposed model can achieve a excellent balance between model parameters and performance.

**Evaluation Results on CamVid Dataset.**

Table 3: Performance comparison of our proposed ECMNet and other representative methods on the CamVid dataset.

| Method | Year | Resolution | Backbone | Parameter (M)↓ | Speed (FPS)↓ | mIoU (%)↑ |
|---|---|---|---|---|---|---|
| ENet(Paszke et al., 2016) | 2016 | $360 \times 480$ | No | 0.36 | 68 | 51.3 |
| SegNet(Badrinarayanan et al., 2017) | 2017 | $360 \times 480$ | VGG-16 | 29.45 | 87 | 55.6 |
| NDNet(Yang et al., 2020) | 2021 | $360 \times 480$ | No | 0.50 | 78 | 57.2 |
| DFANet(Li et al., 2019b) | 2019 | $360 \times 480$ | Xception | 7.80 | 120 | 64.7 |
| DABNet(Li et al., 2019a) | 2019 | $360 \times 480$ | No | 0.76 | 136 | 66.4 |
| FDDWNet(Liu et al., 2020) | 2020 | $360 \times 480$ | No | 0.80 | 79 | 66.9 |
| ICNet(Zhao et al., 2018) | 2018 | $720 \times 960$ | PSPNet-50 | 26.50 | 28 | 67.1 |
| FBSNet(Gao et al., 2022) | 2023 | $360 \times 480$ | No | 0.62 | 120 | 68.9 |
| MSCFNet(Gao et al., 2021) | 2022 | $360 \times 480$ | No | 1.15 | 110 | 69.3 |
| LETNet(Xu et al., 2023) | 2023 | $360 \times 480$ | No | 0.95 | 21 | 70.5 |
| HAFormer(Xu et al., 2024a) | 2024 | $360 \times 480$ | No | 0.60 | 118 | 71.1 |
| MGSeg(He et al., 2021) | 2021 | $736 \times 736$ | ResNet-18 | 13.3 | 127 | 72.7 |
| **ECMNet (Ours)** | - | $360 \times 480$ | No | 0.87 | 55 | **73.6** |

The gray box denotes the best value of the current metric.

As shown in Table 3, to further verify the effectiveness and generalization capacity of our proposed ECMNet, we conducted comparative experments with our method and other lightweight models on the CamVid dataset. Obviously, the MGSeg(He et al., 2021) just achieved the 72.7% mIoU with 13.3M model parameters which is lower performance and larger model parameters compared our proposed method. Therefore, our method has achieved the best accuracy by only using 0.87M parameters. Compared to Cityscapes, the higher overall performance on the CamVid dataset is due to our designed modules and strategies, which better capture feature of small size datasets. Per-class results are detailed in Table 4 of Appendix further demonstrate the advantages of our proposed ECMNet.

## 5 CONCLUSION

In this study, we proposed a lightweight semantic segmentation network that combines Mamba and Convolutional Neural Networks (CNNs). We fused the local feature extraction capability of convolutional neural networks with long-range dependencies of Mamba to model. Specifically, we introduced a Feature Fusion Module (FFM) as a capsule-based framework in the middle of the model, which can better capture global feature information. Additionally, an Enhanced Dual Attention Module (EDAB) designed in the convolutional neural network learned more local feature information while ensuring simplicity and lightweight. Meanwhile, in order to compensate for the local feature information lost by CNNs, multi-scale long connections are utilized in the model. Moreover, We design a Multi-Scale Attention Unit (MSAU) for cross-layer connections, effectively boosting discriminative features and attenuating noise. Extensive experimental results demonstrate that our proposed model achieves an excellent balance between model scale and performance.

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

# A   APPENDIX

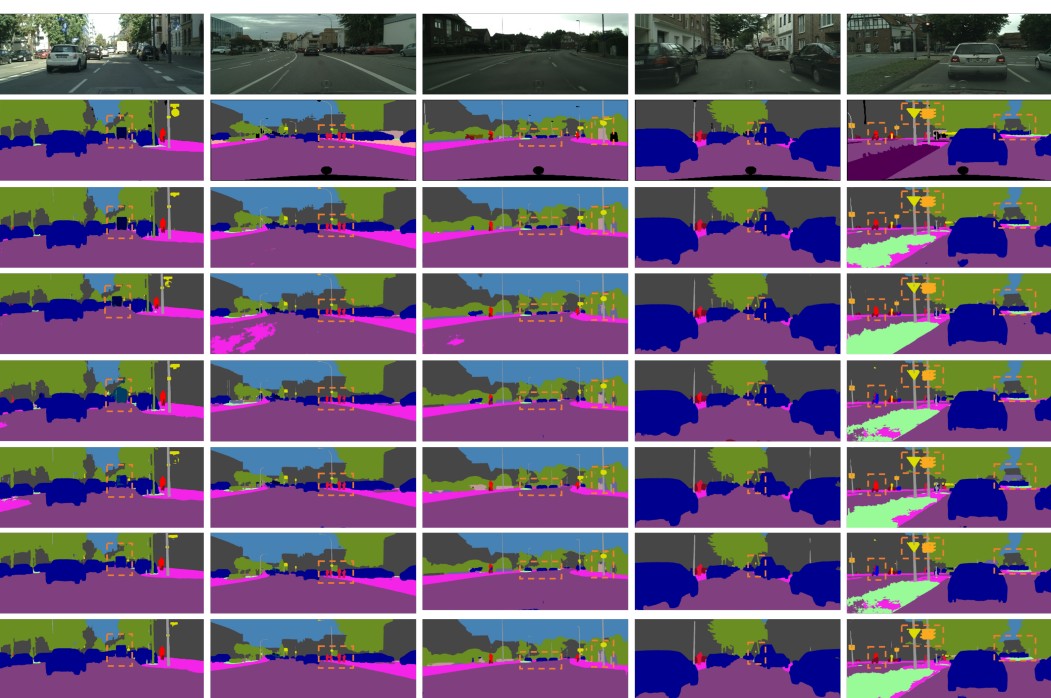

Figure 7: Visualization outcomes on Cityscapes dataset. From top to bottom: original input images, ground truths, predictions of ECMNet, LETNet(Xu et al., 2023), LEDNet(Wang et al., 2019), ERFNet(Romera et al., 2017), ESPNet(Mehta et al., 2018) and ENet(Paszke et al., 2016). Note that five examples are shown. Starting with the third line, each line has rectangular dashed boxes representing the segmentation results where a distinction exists.

Table 4: Performance comparison of our proposed ECMNet and the state-of-arts lightweight methods about per-class results on the CamVid dataset.

| Method | Roa | Sid | Bui | Wal | Fen | Pol | TLi | TSi | Veg | Ter | Sky | Ped | Rid | Car | Tru | Bus | Tra | Mot | Bic | mIoU(%) |
|---|---|---|---|---|---|---|---|---|---|---|---|---|---|---|---|---|---|---|---|---|
| EnetPaszke et al. (2016) | 96.3 | 74.2 | 75.0 | 32.2 | 33.2 | 43.4 | 34.1 | 44.0 | 88.6 | 61.4 | 90.6 | 65.5 | 38.4 | 90.6 | 36.9 | 50.5 | 48.1 | 38.8 | 55.4 | 58.3 |
| ESPNetMehta et al. (2018) | 97.0 | 77.5 | 76.2 | 35.0 | 36.1 | 45.0 | 35.6 | 46.3 | 90.8 | 63.2 | 92.6 | 67.0 | 40.9 | 92.3 | 38.1 | 52.5 | 50.1 | 41.8 | 57.2 | 60.3 |
| CGNetWu et al. (2020) | 95.5 | 78.7 | 88.1 | 40.0 | 43.0 | 54.1 | 59.8 | 63.9 | 89.6 | 67.6 | 92.9 | 74.9 | 54.9 | 90.2 | 44.1 | 59.5 | 25.2 | 47.3 | 60.2 | 64.8 |
| ESPNet-v2Lin et al. (2023) | 97.3 | 78.6 | 88.8 | 43.5 | 42.1 | 49.3 | 52.6 | 60.0 | 90.5 | 66.8 | 93.3 | 72.9 | 53.1 | 91.8 | 53.0 | 65.9 | 53.2 | 44.2 | 59.9 | 66.2 |
| ERFNetRomera et al. (2017) | 97.7 | 81.0 | 89.8 | 42.5 | 48.0 | 56.3 | 59.8 | 65.3 | 91.4 | 68.2 | 94.2 | 76.8 | 57.1 | 92.8 | 50.8 | 60.1 | 51.8 | 47.3 | 61.7 | 68.0 |
| DABNetLi et al. (2019a) | 96.8 | 78.5 | 90.9 | 45.4 | 50.2 | 59.1 | 65.2 | 70.8 | 92.5 | 68.2 | 94.6 | 80.5 | 58.5 | 92.7 | 52.7 | 67.2 | 50.9 | 50.4 | 65.7 | 70.0 |
| CFPNetDing et al. (2024) | 97.8 | 81.4 | 90.5 | 46.4 | 50.6 | 56.4 | 61.5 | 67.7 | 92.1 | 68.9 | 94.3 | 80.4 | 60.7 | 93.9 | 51.4 | 68.0 | 50.8 | 51.2 | 67.7 | 70.1 |
| LEDNetWang et al. (2019) | 98.1 | 79.5 | 91.6 | 47.7 | 49.9 | 62.8 | 61.3 | 72.8 | 92.6 | 61.2 | 94.9 | 76.2 | 53.7 | 90.9 | 64.4 | 64.0 | 52.7 | 44.4 | 71.6 | 70.6 |
| **ECMNet (Ours)** | 97.1 | 80.8 | 90.9 | 44.2 | 53.4 | 60.8 | 61.9 | 72.4 | 91.7 | 60.4 | 93.9 | 75.2 | 52.7 | 92.0 | 65.3 | 76.5 | 66.5 | 37.8 | 69.1 | **70.6** |

The gray box denotes the best mIoU of the current class. Roa, Sid, Bui, Wal, Fen, Pol, TLi, TSi, Veg, Ter, Sky, Ped, Rid, Car, Tru, Mot and Bic reprsent Road, Sidewalk, Building, Wall, Fence, Pole, Traffic Light, Traffic Sign, Vegtation, Terrain, Sky, Pedestrain, Rider, Car, Truck, Motorcycle and Bicycle respectively.

