# OpenReview forum: "ECMNet: Lightweight Semantic Segmentation  with Efficient CNN-Mamba Network"
_ICLR.cc/2026/Conference — ICLR 2026 Conference Withdrawn Submission_

### Official Review · Reviewer_iMoH · 2025-10-27

**Soundness:** 2
**Presentation:** 2
**Contribution:** 2
**Rating:** 4
**Confidence:** 4

**Summary:**

The paper proposed a lightweight ECMNet for semantic segmentation, which effectively combines CNN and Mamba in a framework to leverage their complementary strengths in local detail capture and long-range dependency modeling. It introduces three key modules: an EDAB for lightweight feature refinement, a MSAU for spatial–channel aggregation, and a FFM for hierarchical feature integration. Experimental results on Cityscapes and CamVid show that ECMNet achieves 70.6% and 73.6% mIoU, respectively, with only 0.87M parameters and 8.27G FLOPs, demonstrating an balance between segmentation accuracy and computational efficiency.

**Strengths:**

First, it effectively integrates CNN and Mamba to balance local detail capture with long-range dependency modeling, addressing the global context limitation common in semantic segmentation.
Second, it achieves an excellent accuracy–efficiency trade-off, reaching 70.6% mIoU on Cityscapes and 73.6% mIoU on CamVid with only 0.87M parameters and 8.27G FLOPs, demonstrating strong computational efficiency.

**Weaknesses:**

Incremental novelty: The core contribution is a module combination (CNN + Mamba + attention/fusion blocks) rather than a clear algorithmic breakthrough.
Insufficient breadth of evaluation: Results are limited to Cityscapes and CamVid without robustness tests (domain shift, noise, resolution), or comparisons against the most recent lightweight/SSM-based segmenters.

**Questions:**

1.The mathematical formulas (e.g., Equations 1–9) are densely stacked, which affects readability and clarity.
2.The innovation is limited. The proposed network is essentially a combination of CNN and Mamba, where the EDAB module resembles the attention mechanism used in Transformers, and the MASU module is similar to CBAM, making the contribution incremental rather than fundamentally novel.
3.The manuscript lacks a clear description of the loss function and evaluation metrics, which are necessary for reproducibility.
4.The ablation study does not include qualitative comparisons, making it difficult to visually assess the impact of each component.
5.The CamVid dataset lacks qualitative visual comparisons, and in Figure 7 (Cityscapes dataset), recent methods from the past two years are also missing.
6.In Tables 2 and 3, the comparisons with recent methods are insufficient. Table 2 includes only two 2024 methods, and Table 3 includes only one 2024 method.
7.The paper lacks any robustness analysis or validation, such as testing under noise, occlusion, or domain variation.
8.In Table 4, the proposed method performs similarly to the 2019 method LEDNet; the reason for this comparable performance should be clearly explained.

**Details Of Ethics Concerns:**

1.The mathematical formulas (e.g., Equations 1–9) are densely stacked, which affects readability and clarity.
2.The innovation is limited. The proposed network is essentially a combination of CNN and Mamba, where the EDAB module resembles the attention mechanism used in Transformers, and the MASU module is similar to CBAM, making the contribution incremental rather than fundamentally novel.
3.The manuscript lacks a clear description of the loss function and evaluation metrics, which are necessary for reproducibility.
4.The ablation study does not include qualitative comparisons, making it difficult to visually assess the impact of each component.
5.The CamVid dataset lacks qualitative visual comparisons, and in Figure 7 (Cityscapes dataset), recent methods from the past two years are also missing.
6.In Tables 2 and 3, the comparisons with recent methods are insufficient. Table 2 includes only two 2024 methods, and Table 3 includes only one 2024 method.
7.The paper lacks any robustness analysis or validation, such as testing under noise, occlusion, or domain variation.
8.In Table 4, the proposed method performs similarly to the 2019 method LEDNet; the reason for this comparable performance should be clearly explained.

---

### Official Review · Reviewer_benL · 2025-10-29

**Soundness:** 2
**Presentation:** 3
**Contribution:** 2
**Rating:** 2
**Confidence:** 2

**Summary:**

This paper introduces ECMNet, a lightweight network for semantic segmentation that aims to efficiently combine Convolutional Neural Networks (CNNs) and the Mamba architecture. This paper employs a U-Net-like encoder-decoder framework. The main building blocks of the encoder and decoder are a "Enhanced Dual-Attention Block" (EDAB). To improve the fusion of features across different scales, a "Multi-Scale Attention Unit" (MSAU) is integrated into the skip connections. The core novelty claim lies in the use of a Mamba-based "Feature Fusion Module" (FFM) at the deepest bottleneck of the network to capture global long-range dependencies. Experiments on the Cityscapes and CamVid datasets show that ECMNet achieves competitive performance with a very low parameter count (0.87M), demonstrating a strong balance between accuracy and efficiency.

**Strengths:**

The authors have conducted comprehensive ablation studies (Table 1) that demonstrate the contribution of each proposed component (the long connections, the MSAU, and the FFM). This rigorous experimental approach is commendable and effectively validates the efficacy of their design choices.

**Weaknesses:**

1.The central concept of combining a CNN backbone with a module designed for long-range dependency modeling (in this case, Mamba, previously Transformers) has become a very common, almost standard, paradigm in semantic segmentation and other vision tasks. The hybrid architecture itself is no longer a groundbreaking contribution. While applying Mamba is a timely and logical step, the paper treats it more like a plug-and-play replacement for self-attention, without providing a deeper investigation into the fundamental principles or unique synergies of combining Mamba's state-space modeling with CNN's inductive biases. This feels more like an expected, incremental evolution rather than a novel research direction.

2.The paper's novelty is further weakened by its module design, which appears to be more of an engineering amalgamation of existing techniques than an approach driven by a novel, unifying principle. The MSAU employs a conventional multi-scale parallel-branch design reminiscent of Inception, combined with a standard channel attention mechanism; its contribution lies in the arrangement, not a new paradigm.  The use of Mamba in the FFM represents its most straightforward application—placing it only at the bottleneck. The work does not explore alternative integration strategies or provide a compelling argument for why this specific placement is optimal.

**Questions:**

None

---

### Official Review · Reviewer_uHTW · 2025-10-30

**Soundness:** 2
**Presentation:** 2
**Contribution:** 2
**Rating:** 4
**Confidence:** 4

**Summary:**

This paper proposes EMCNet, a lightweight semantic segmentation model that integrates CNN and Mamba architectures. A notable strength of the work is its efficient design, achieving a low parameter count of only 0.87M parameters and being trained from scratch. However, the paper could be significantly strengthened by addressing two main concerns: the model's performance, which is not state-of-the-art and appears modest by current standards, and the completeness of the experimental evaluation, which requires further validation to robustly demonstrate the method's effectiveness.

**Strengths:**

- By integrating CNN and Mamba within a lightweight architecture, the paper proposes ECMNet, a model with a total of 0.87M parameters. This is a very competitive parameter count compared to existing segmentation models.
- The paper is well-written and well-structured. The presentation is clear and easy to follow.

**Weaknesses:**

- The figures in the paper are not in vector format, which causes blurriness when zoomed in. Replacing them with vector graphics would improve the visual quality.
- The experimental results do not sufficiently demonstrate the effectiveness of the proposed EMCNet. As shown in Table 2, LETNet from 2023 with similarly fewer than 1M parameters achieves a higher mIoU (+2.2) and faster inference speed (+107 FPS). Furthermore, the performance of EMCNet is only comparable to that of LEDNet from 2019. Given the current state of the art in 2025, the performance of EMCNet is relatively low, which undermines the authors' claim regarding the superiority or competitiveness of their method.
- While the authors propose three modules and claim their respective functions, the experimental evidence is insufficient to substantiate these claims. For instance, in line 216, it is claimed that the two branches of the EDAB module are responsible for capturing local and global information respectively. However, no experimental validation (such as feature visualization or ablation studies isolating each branch's effect) is provided to support this key design motivation. Similar issues exist for the other modules. Without such evidence, the claims about the mechanisms behind the modules' effectiveness remain unverified.
- The paper's claim of novelty is limited. While the authors propose using an SS2D block to capture long-range dependencies, this approach is not novel, as it was already employed by the SegMAN model (accepted to CVPR 2025) for semantic segmentation. Therefore, this work cannot be considered the first to apply a Mamba-based architecture (via SS2D) to this task. The authors need to more precisely articulate the actual incremental contribution of EMCNet relative to existing contemporaneous works like SegMAN.
- Typo in Line 213: Missing space after comma.

**Questions:**

The paper positions the work as a combination of CNN and Mamba architectures. However, the proposed EDAB module also incorporates an attention mechanism. This inclusion suggests that the network is more accurately described as a hybrid of CNN, Mamba, and Attention. The authors should reconsider the title and the core definition of their work to ensure they precisely reflect all key architectural components, avoiding potential misunderstanding.

---

### Official Review · Reviewer_hizf · 2025-10-31

**Soundness:** 3
**Presentation:** 3
**Contribution:** 2
**Rating:** 4
**Confidence:** 4

**Summary:**

This paper proposes a lightweight semantic segmentation model called ECMNet that combines CNN and Mamba, achieving a favorable balance between parameter efficiency and accuracy on the Cityscapes and CamVid datasets.

**Strengths:**

The method achieves a well-balanced parameter-accuracy trade-off by leveraging existing technologies, demonstrating tangible practical value for real-world applications.

**Weaknesses:**

1. The novelty of the proposed method appears incremental, as hybrid architectures like CNN-Mamba, CNN-Transformer, and Transformer-Mamba have been extensively studied. The authors do not provide a compelling justification for their specific design. The manuscript would be significantly strengthened by a thorough discussion addressing: 1) The distinct advantages of the proposed CNN-Mamba network over existing hybrid paradigms. 2) How the architecture is specifically tailored for the demands of semantic segmentation. 3) A precise analysis of its contributions to model lightweighting.

2. The experimental comparisons in this study lack timeliness, as they are primarily confined to methods established in 2021 or earlier. To properly validate the competitiveness of the proposed approach, it is essential to include comparisons with more recent state-of-the-art methods. Updating the benchmarks would provide a more convincing and contemporary assessment of the method's performance.

**Questions:**

1. What are the specific unsolved challenges addressed in this manuscript? While combining CNN's local feature modeling with Mamba's long-range dependency strengths achieves accuracy-efficiency balance, this theoretical foundation has been extensively validated in prior works (e.g., HResFormer, PFormer, DMFC-UFormer). Merely restating this synergy does not suffice as a research motivation.

2. The MSAU module is described as a "multi-scale attention unit," but the core differences between it and classical multi-scale attention (e.g., PSPNet, ASPP) are not elaborated. Does MSAU introduce new parameter optimization strategies? What are its differences from multi-path dilated convolution?

---

### Note · Authors · 2025-11-12

I have read and agree with the venue's withdrawal policy on behalf of myself and my co-authors.